# GeoVeX: Geospatial Vectors with Hexagonal Convolutional Autoencoders

## Abstract

We introduce a new geospatial representation model called GeoVeX to learn global vectors for all geographical locations on Earth land cover. GeoVeX is built on a novel model architecture named Hexagonal Convolutional Autoencoders (HCAE) combined with a Zero-Inflated Poisson (ZIP) reconstruction layer, applied to a grid of Uber's H3 hexagons, each one described by the histogram of OpenStreetMap (OSM) geographical tags occurrences. GeoVeX is novel on two aspects: first, it produces pre-trained task-agnostic geospatial vectors with H3 and OSM that are, for the first time, contextualized on the neighboring hexagons features, by leveraging an hexagonal convolutional autoencoder applied on an H3/OSM grid centered on the location to embed; secondly, it introduces a zero-inflated Poisson autoencoder reconstruction layer, to adapt a standard autoencoder network to train on sparse geographical count data distributed on an hexagonal grid. Experiments demonstrate that GeoVeX embeddings improve upon two state-of-the-art geospatial location representations models, Hex2Vec and Space2Vec, on two different downstream tasks: worldwide listings price prediction in the travel industry, and hyperlocal interpolation of climate data from weather stations. The qualitative analysis of the latent representation structures learnt by GeoVeX showcases the higher quality of the geographical structures learnt by the geographically contextualized embeddings learnt by GeoVeX.

## 1    Introduction

Entity embedding is ubiquitous in a variety of Machine Learning tasks thanks to its many advantages: it captures the semantics of each entity in the context of a given domain; it enables transfer learning to different related tasks; it reduces the sparsity of the entity representation and compresses the feature space. In NLP domain, global word embedding models, such as *Word2Vec* (Mikolov et al., 2013), *GloVe* (Pennington et al., 2014) and *BERT* (Devlin et al., 2019) have been successful at capturing the word semantics of big open-source vocabularies (e.g. Wikipedia, Gigaword) and are used to transfer learning to multiple downstream tasks, such as sentiment analysis (Tang et al., 2014; Deho et al., 2018; Alamoudi & Alghamdi, 2021), question retrieval (Zhou et al., 2015), and medical semantics (Wang et al., 2018). Similar approaches inspired by NLP have been since then proved to be useful in many industrial domains, where multiple models have been proposed for learning the latent representations of entities specific to an industry, such as *Product2Vec* (Biswas et al., 2017) and *User2Vec* (Hallac et al., 2019) in e-commerce, or *Wave2Vec* (Baevski et al., 2020) in speech representation, just to name a few.

In comparison, in the field of Geographic Information Science (GIS), a global set of task-agnostic embeddings for geographical space representation can benefit multiple domains and use cases, such as: price prediction for houses (Wang et al., 2021), hotel rooms (Kisilevich et al., 2013), and vacation homes (Islam et al., 2022; Pradip & Suthar, 2022); interpolation of climate variables such as temperature and pressure (Wu & Li, 2013); computer vision tasks with geo-located images (Berg et al., 2014). These tasks, just to name a few, have in common the application of some transformations to the spatial coordinates, but they do not leverage the spatial distribution of geo entities (such as parks, water, beach, buildings, streets, bars, etc.), which convey a more rich information of the geographical context. Besides, in terms of modelling, previous approaches to learn geospatial embeddings have a set of limitations, such as being non-contextual, task-specific and/or region-specific (Sec. 2) that we address with a novel model architecture and loss function formulation (Sec. 3.6).

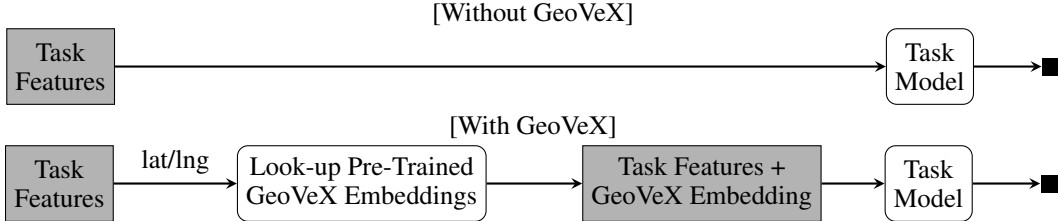

Figure 1: Workflow to use GeoVeX embeddings in downstream tasks: the task features are expanded by simply concatenating the GeoVeX embedding associated to the H3 hexagon corresponding to the latitude (lat) and longitude (lng) coordinates of each task item, without retraining the embeddings.

Our new approach to learn the geospatial embedding of each location on Earth, based on nearby geographical entities, aims to pre-train a finite set of embeddings that can cover the whole Earth, so that they can be stored and used with ease as extra features in any Machine Learning task where entities have latitude and longitude coordinates. The general workflow is summarized in Fig. 1. To achieve the goal of wide adoption to multiple downstream tasks, we leverage Uber Hexagonal Hierarchical Spatial Index grid system named H3[1] to spatially index the data coordinates into small regions of approximately the same size, since H3 minimizes map distortion. A pair of coordinates $(i, j)$ is thus represented by a unique H3 hexagonal id for which we learn a GeoVeX embedding.

To learn GeoVeX embeddings that have a geographical semantic, we associate each H3 hexagon to the geographical tags of the entities obtained from OpenStreetMap (OSM)[2]. OSM is a project that creates and distributes free worldwide geographic data, and, as of January 2022, has ≈7 billion nodes and ≈4 million map changes per day. This makes it the equivalent of Wikipedia for word embeddings: a massive, scalable, and information-rich global open dataset for creating and updating global embeddings. In particular, OSM contains nodes, ways and relations, which together can be transformed to points, lines and polygons, each one characterized by a set of semantic tags, such as *amenity:bar*, *highway:motorway*, *natural:forest*. By intersecting these OSM entities with the H3 hexagons, and by using a Bag-Of-Words (BOW) model on their tags, each hexagon can be sparsely described by a $K$-dimensional histogram vector, where $K$ is the size of a subset of the vocabulary of OSM geographical tags, where each element represents the number of times an entity with the respective tag is contained or intersects the hexagon itself. This information needs then to be properly aggregated to produce an embedding while, at the same time, taking into account the information from neighboring hexagons, which provide the geographical context. This concept follows the first law of geography: *everything is related to everything else, but near things are more related than distant things* (Tobler, 1970). The convolution operation, borrowed from Computer Vision, presents in this domain challenges to address: 1) not square, but hexagonal grids, and 2) different distribution of each "channel": not dense pixel values, but highly sparse counts. GeoVeX model aims to bridge the gap of convolutional neural networks usage on these hexagonal grids described by sparse counts.

In summary, the **contributions** of our work are:

1. the GeoVeX architecture design to learn task-agnostic pre-trained location embeddings with H3 and OSM that are for the first time contextualized on the neighboring hexagons. We demonstrate their expressive power qualitatively, by using an analysis of the cosine similarities (Sec. 4.1), and quantitatively, by adding the embeddings to the feature set of two downstream tasks: price prediction in the worldwide travel industry (Sec. 4.2) and temperature interpolation of climate data from weather stations (Sec. A.6);

2. the novel Zero-Inflated Poisson (ZIP) autencoder's probabilistic decoder block, which is trained with a spatial contextual loss function, to adapt the standard reconstruction layer of autoencoders network to the case of zero-inflated spatial contextual count data produced by the H3 grid and the OSM entity tags counts (Sec. 3.6).

---

[1]https://eng.uber.com/h3

[2]https://www.openstreetmap.org/

## 2 RELATED WORK

**Global geospatial embeddings** *Hex2Vec* (Woźniak & Szymański, 2021) and *Space2Vec* (Mai et al., 2020) are the closest approaches to our work, since they create pre-trained location-specific embeddings, which can add geospatial information to any downstream task without any task-specific retraining nor any manual location imputation logic.

Hex2Vec (Woźniak & Szymański, 2021) is a state-of-the-art work that uses H3 hexagons described by OSM entities, and develops an approach to learn geospatial embeddings. GeoVeX follows a similar approach to create the training setup (H3 as grid methodology, and OSM as provider of geographical entities), but it is different in terms of both model design and loss function formulation. In fact, Hex2Vec embeddings are not contextualized, since the embedding of an H3 hexagon does not consider any information of the surrounding hexagons (e.g. an hexagon described with just the tag "street" is always embedded equally worldwide, without any distinction if this hexagon is surrounded by others with tags of "offices" or "ski-lifts" or "beaches"). Regarding the loss, the authors proposed a skip-gram model with negative sampling, paired with a triplet loss function: for each hexagon to embed (the triplet *anchor*), the *positive* hexagon is taken from the first adjacent ring of hexagons, and the *negative* hexagon is sampled from at least 3 rings away from it. It is easy to see that imposition of complete similarity between the hexagon to embed (the anchor) and its adjacent ones (the positives) in the loss formulation is not always appropriate since it imposes perfect similarity of a beach-like hexagon's embedding with a town-like one in case they are adjacent. Finally, the authors applied the approach only to 54 cities, and without a quantitative analysis of the improvements on down-stream tasks.

Space2Vec (Mai et al., 2020) is one of the state-of-the-art methods proposed to encode a pairs of coordinates in a high dimensional embedding which can be used in downstream tasks. In particular it uses trigonometric functions with different frequencies to encode a given position in space instead of using the simple pairs of coordinates in a modeling task. The approach can actually be considered as a worldwide embedding model in the sense that any coordinate on Earth can be transposed to a latent representation. However, the embeddings are not learnt based on the geographical entities (e.g. buildings, streets, bars, etc.), so Space2Vec vectors cannot be really considered as pre-trained embeddings aiming at learning geographic characteristics of a location, in order to draw similarities of different regions on Earth.

**Local or task-specific geospatial embeddings** Other approaches to learn geospatial embeddings in GIS domain are trained for specific tasks and/or specific regions, so they cannot be used as pre-trained geospatial embeddings without task-specific retraining and/or without manual imputation logic. *Region2Vec* (Xiang, 2020) combines Point Of Interest (POI) and mobile sensors data obtained in a specific region in China; *Zone2Vec* (Du et al., 2018) uses trajectories generated by taxis in Beijing. *Urban2Vec* (Wang et al., 2020b) uses Street View images and focuses only on the urban use cases, where such images are available. *RegionEncoder* (Jenkins et al., 2019) combines mobility from taxis, categories of POI and satellite images to create a model specific to two urban regions. *Tile2Vec* (Jean et al., 2019) applies an unsupervised representation learning on satellite images, while using the triplet loss formulation, which leads to the same issues described for Hex2Vec. In (Islam et al., 2022), the authors apply Moran Eigenvector Spatial Filtering (MESF) for the specific task of predicting AirBnB listing price in San Jose County in US. In (Mac Aodha et al., 2019), the location encoder is specifically learnt from photos to improve specific image-classification tasks, and learnt embeddings cover only the locations in the task-specific training dataset. In (Sheehan et al., 2019), the authors learn task-specific location embeddings from geolocated Wikipedia articles, which cover a limited set of coordinates, thus requiring manual imputation logic for unknown locations.

**Convolutional Autoencoders** In terms of network architecture, our design is inspired by recent works on Convolutional Autoencoders (CAE), among which the deep CAE architecture of Guo et al. (2017) represents an example of the state of the art. In addition, (Hahner & Garcke, 2022) recently introduced the use of hexagonal convolutional autoencoders. However, these architectures are designed to work with images, and not with tensors that represent zero-inflated count variables, so we propose some modifications to account for this.

## 3 PROPOSED APPROACH: GEOVEX

In this section, first, we formalize the problem of training geospatial embeddings of a pair of coordinates on Earth, then we present the proposed GeoVeX model architecture, its input format and its novel specific components: hexagonal convolutions, the zero-inflated Poisson reconstruction layer for the autoencoder, and the loss function used to train the network.

### 3.1 PROBLEM FORMALIZATION

Given a pair of latitude and longitude coordinates on Earth $p = (x, y) \in \mathbb{R}^2$, define a function $F_\theta(p) : \mathbb{R}^2 \to \mathbb{R}^D$ parameterized by $\theta$ which maps the point $p$ to a $D$-dimensional embedding $e = F_\theta(p)$, which represents the geographical characteristics of the space nearby the given point proportionally to the distance to it, in accordance to Tobler (1970).

In this work, we propose to formulate the $F$ function as $F_\theta(p) = f_\theta(t(p), \mathcal{N}_{n'}(t(p)))$, where $f_\theta : \mathbb{R}^K \times \mathbb{R}^{n' \times K} \to \mathbb{R}^D$ is the GeoVeX encoder, and $t : \mathbb{R}^2 \to \mathbb{R}^K$ is the function which assigns the vector of counts $z = t(p)$ of $K$ geographical tags of the entities that intersect the hexagon where the point $p$ resides, and $\mathcal{N}_{n'}(t(p))$ is the set of count vectors of $n'$ neighbor hexagons of $p$. The GeoVeX pre-trained embedding $e$ can later be concatenated to existing features of any downstream task involving entities described by latitude and longitude coordinates. This effectively serves to expand the features set used by the model addressing the machine learning task to solve.

### 3.2 GEOVEX MODEL 3D TENSOR INPUT FROM 2D HEXAGONAL CELLS

In this section we briefly describe the GeoVeX 3D input tensor required by the network architecture. GeoVeX uses an hexagonal tiling system, which is the state-of-the-art representation of a geographical grid, because it guarantees the isotropy of local neighbourhoods (Wang et al., 2020a). By analogy with image domain, in GeoVeX the first 2 dimensions represent the 2D coordinates of the hexagon in a spatial coordinate system and the 3rd dimension describes the count of $K$ OSM tags of the entities intersecting a given H3 cell, following the Bag-Of-Words (BOW) model. After cleaning and preprocessing, we cut the number of tags to the top $K = 1024$ most frequent ones with worldwide minimum coverage, since the vocabulary presents a long tail distribution. More details can be found in Section A.3 of Appendix.

Each hexagon's neighborhood is defined by the $r$-rings neighboring cells as described in Fig. 2 for $r = 7$ rings around the point $(0, 0)$ ($r = 7$ has been experimentally defined to fit our GPU memory constraints). The importance of each ring decays with distance to center, in alignment with the first law of geography (Tobler, 1970). This is implemented using a distance weighting kernel in the model loss function formulation in Sec. 3.6, so that the representation of an hexagon is influenced by neighboring hexagons proportionally to the hexagonal distance to it, making them contextualized. This is not the case for Hex2Vec model design, instead.

Transposition of the 2D hexagonal grid to 2D matricial form, for the first 2 dimensions of the 3D-tensor, is done via *axial* coordinate system representation (see Luo et al. (2019) for more details on different transposition methods), and is depicted in Fig. 3. The final 3D representation is $(2r + 1) \times (2r + 1) \times K$. To enable correct convolutions within this network design, the dimensions of the input matrix are required to be a power of 2, which can be achieved through padding. Since we trained the network with $r = 7$ the input of the GeoVeX model is a $16 \times 16 \times 1024$ 3D tensor.

### 3.3 MODEL ARCHITECTURE

The architecture of GeoVeX network is summarized in Fig. 4. The base design is the one of a Convolutional AutoEncoders (CAE) network (Guo et al., 2017), which is composed of two main components: the encoder $h = f(z)$ and the decoder $z' = g(h)$, where $z$ is the input tensor, $h$ is its latent representation learnt by applying the encoder block, and $z'$ is the reconstructed tensor created by the decoder block. The parameters of the CAE network are updated by minimizing the reconstruction error described in Sec. 3.6.

The encoder is a set of convolutional blocks stacked on the 3D input tensor to extract hierarchical features. The output units of the convolutions are flattened to form a vector, and are passed to a

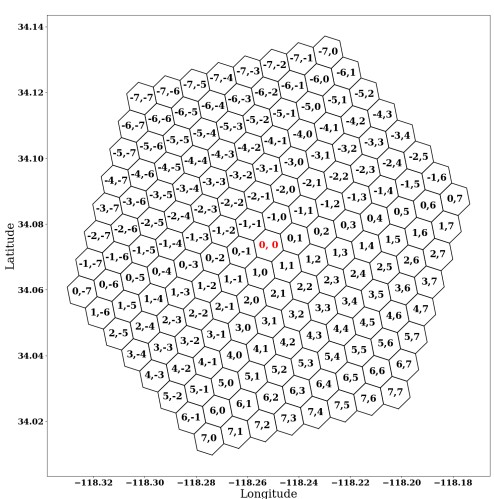 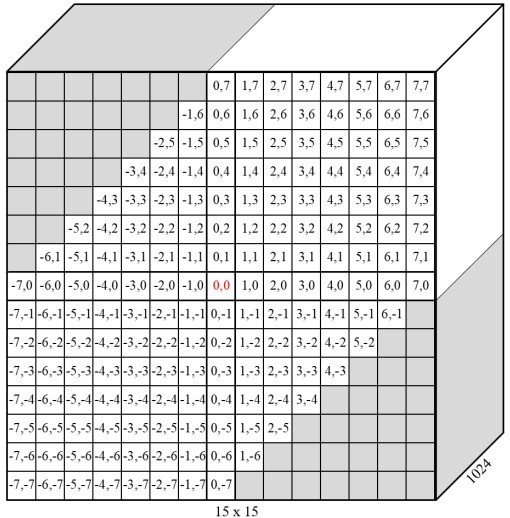

Figure 2: The hexagonal grid of the $r$-rings centered on the hexagon to embed, with the axial coordinates $(i, j)$ assigned to each hexagon based on its position in the grid.

Figure 3: The transposed hexagonal grid represented as a 3D matrix, using the axial coordinates transposition and assigning the OSM counts as 1024 tensor channels. In grey, the padding cells that are not valid.

fully connected layer with $D$ units, the final dimension of GeoVeX embedding. In practice, we have found that $D = 32$ is a good trade-off between expressiveness and cost to store 200+ millions vectors worldwide. The input tensor is thus transformed by the encoder into a 32-unit latent embedding representation $h$, forcing the autoencoder to capture the most salient features that minimize the loss function. The decoder is defined as a set of convolutional transposed layers to transform embedded feature back to original input, followed by a novel Zero-Inflated Poisson layer of our contribution to reconstruct the count tensor in input (Sec. 3.5). Batch Normalization (Ioffe & Szegedy, 2015) is used to re-center and re-scale the counts tensor in input and output of every convolution operation. The usage of Batch Normalization and ReLU is aligned to (Ioffe & Szegedy, 2015): we apply the normalization before the non-linearity and we remove the redundant bias term in the convolution operation. In terms of up/sub sampling, GeoVeX uses convolutional layers with stride of 2, instead of convolutional layers followed by pooling layers (as in (Hahner & Garcke, 2022)), in both the encoder (standard convolution) and the decoder (transposed convolution, sometimes referred as "deconvolution"), since Springenberg et al. (2014) demonstrated that replacing all the pooling operations in a network with strided-convolutions improves overall performance.

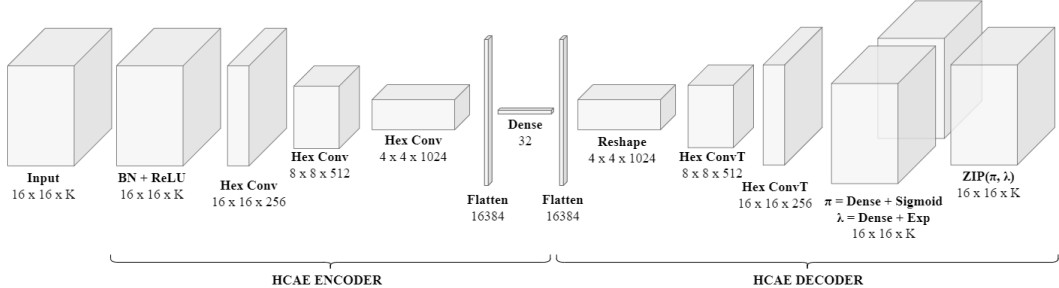

Figure 4: Illustration of the GeoVeX network architecture, showing the different layers of the Hexagonal Convolutional Autoencoders (HCAE) neural network with a Zero-Inflated Poisson (ZIP) probabilistic head layer to reconstruct the hexagonal matrix with geographical entity tags count. Each hexagonal convolution block and hexagonal transposed convolution block is composed by the respective convolution operator followed by Batch Normalization and ReLU operations.

### 3.4 HEXAGONAL CONVOLUTIONS

The main building block of both the encoder and the decoder is the hexagonal convolution operator, which aims at extracting relevant geographical features at different granularity. The network assumes in input an hexagonal grid transposed to a two dimensional matrix using the the *axial* coordinate system representation as described in Sec. 3.2. In this representation, unused matrix cells are put in the corners, avoiding empty cells in the middle of the matrix and supporting the hexagonal convolutional autoencoder network described below. In this context, it is shown that hexagonal convolutions give better results than standard convolutions in case of hexagonal shaped data (Steppa & Holch, 2019; Hoogeboom et al., 2018). In particular, the hexagonal convolution is achieved by multiplying the 3x3 convolution kernel with a 3x3 mask to set to zero the kernel at top-left and bottom-right cells (the mathematical formula of the 3x3 masked hexagonal convolution is given in Section A.1 of Appendix).

### 3.5 ZERO-INFLATED POISSON RECONSTRUCTION LAYER

Since the network is intended to be used with geographical count data assigned to each hexagon, the Poisson head layer is needed since the variable to reconstruct is coming from a counting process. Moreover, it expects a large number of zeros, since it is evident that for any given hexagon on Earth, only few geographical tags from the full vocabulary of tags are present: this hypothesis shall be verified in the data using the respective tests (Yang et al., 2010; Van den Broek, 1995). Under this scenario, multiple studies have demonstrated that the best fit to such data with excess of zeros is provided by zero-inflated distributions (Ridout et al., 1998; Beckett et al., 2014; Lambert, 1992; Unhapipat et al., 2018).

The Zero-Inflated Poisson (ZIP), which is denoted by $ZIP(\pi, \lambda)$, has two parameters $\pi$ and $\lambda$, and has the following probability mass function, where $0 \leq \pi \leq 1$ and $\lambda \geq 0$:

$$\forall i \in \{1, \ldots, K\}, \mathbb{P}(z_{.,.,i} = c) = \begin{cases} \pi + (1 - \pi) \cdot e^{-\lambda} & \text{if } c = 0 \\ (1 - \pi) \cdot e^{-\lambda} \cdot \frac{\lambda^c}{c!} & \text{if } c \in \{1, 2, 3, \ldots\} \end{cases} \quad (1)$$

In the GeoVeX model, the parameters $\pi$ and $\lambda$ are estimated in the last layer of the decoder, and are used by the above ZIP mixture model to reconstruct, from the H3 embedding, the OSM count of each geographical tag $k$ of each hexagon at axial coordinates $i, j$. In fact, the final decoder layer produces two output variables for each coordinate $i, j, k$: $g_\pi$ and $g_\lambda$. The first one is activated with a *sigmoid* function to produce $\pi$ (Lambert, 1992), and the second one is activated with the *exponential* function to ensure that $\lambda > 0$, as is obviously required:

$$\widetilde{\pi}_{i,j,k} = \frac{e^{g_{\pi|i,j,k}(h)}}{1 + e^{g_{\pi|i,j,k}(h)}} \quad (2)$$

$$\widetilde{\lambda}_{i,j,k} = e^{g_{\lambda|i,j,k}(h)} \quad (3)$$

where $g_{\pi|i,j,k}(h), g_{\lambda|i,j,k}(h)$ are the two output layers of the decoder network which are conditioned on embedding $h$ and produce the estimated $\widetilde{\pi}$ and $\widetilde{\lambda}$ for each coordinate $i, j, k$ (Fig. 4).

### 3.6 GEOVEX LOSS FUNCTION

GeoVeX adapts the Negative Log-Likelihood (NLL) loss function in the context of a hexagonal grid to align with the design of the input data. In fact, the loss formulation encourages first the learning (and, thus, the reconstruction) of the geographical tags count describing the hexagon to embed, and then decay the importance of learning (and, thus, reconstructing) the tags count in each external ring of hexagons, by weighting the loss of each hexagon based on its position in the grid, thus justified simply by grid construction.

In details, the formulation of the GeoVeX loss function is an adaptation of the NLL loss with Zero-Inflated Poisson distribution, in the context of a spatial relationship among the neighbouring regions. Based on the ZIP mass function (Eq. 1), and given that we have $N$ samples in each learning batch, the log-likelihood function of each element of the 3D reconstructed tensor can be derived as:

$$\log \mathcal{L}(\widetilde{\pi}, \widetilde{\lambda}; y)_{i,j,k} = \sum_{n=1}^{N} \mathbb{I}(y_{n|i,j,k} = 0) \ln[\widetilde{\pi}_{n|i,j,k} + (1 - \widetilde{\pi}_{n|i,j,k})e^{\widetilde{\lambda}_{n|i,j,k}}]$$

$$+ \sum_{n=1}^{N} \mathbb{I}(y_{n|i,j,k} > 0)[\ln(1 - \widetilde{\pi}_{n|i,j,k}) - \widetilde{\lambda}_{n|i,j,k} + y_{n|i,j,k} \ln \widetilde{\lambda}_{n|i,j,k} - \ln(y_{n|i,j,k}!)]$$

$$(4)$$

where $n$ is the n-th sample, $i$ and $j$ are the two axial coordinates, $k$ is the $k$-th geographical tag count, and $\mathbb{I}(y_{n|i,j,k})$ denotes an indicator variable that takes value 1 if the condition is verified, and zero otherwise. See derivation in (Hossain & Howlader, 2015) and (Lambert, 1992). To make the loss of each grid hexagon weighting proportional to the ring where it is located, we apply two positional weights to the ZIP NLL loss of each hexagon of the grid, based on its $i, j$ axial coordinates:

- the *distance weighting kernel* $W_{dist}$ decays the importance of each subsequent neighbouring $r$-ith ring of hexagons, based on the inverse of the distance from the central hexagon to the ring number $r$ (i.e. ring $r = 1$ applies a weight of 1/2, while ring $r = 7$ applies a weight of 1/8);

- the *numerosity weighting kernel* $W_{num}$ equalizes the importance of each hexagonal $r$-ith ring, by subdividing the loss among the hexagons that compose it, so that each ring has same total importance in the final loss. By construction, each ring is composed of 6 times the number of the ring number $r$ (i.e. ring $r = 1$ has 6 hexagons, while ring $r = 7$ has 42 hexagons, so each hexagon has to weight $6 \cdot r$ to make each ring equal in importance)

The mathematical formulations in Eq. 5 and 6 are justified by construction (based on the semantic imposed by using an hexagonal grid centered on the hexagon to embed), where $r$ is the hexagonal ring number, and $|i - j| \leq R$ is a simple condition to explicitly put to zero the loss at the invalid top-left and bottom-right values in the axial coordinates matrix representation.

$$w_{dist_{i,j}} = \begin{cases} 1 & \text{if } i = j = 0 \\ 1/(1 + r_{i,j}) & \text{if } |i - j| \leq R \\ 0 & \text{otherwise} \end{cases} \tag{5}$$

$$w_{num_{i,j}} = \begin{cases} 1 & \text{if } i = j = 0 \\ 1/(6 \cdot r_{i,j}) & \text{if } |i - j| \leq R \\ 0 & \text{otherwise} \end{cases} \tag{6}$$

The two loss weighting kernels are depicted in Fig. 6 and 7 in Sec. A.2 of Appendix, to help visualization of the kernel effect. By combining the two weighting kernels, the loss to minimize is the sum of the ZIP negative log-likelihood loss of each $k$-th geographical tag count of each $i, j$ element of the hexagonal grid (Eq. 4), weighted based on its grid position using the kernels $W_{dist}$ and $W_{num}$:

$$L = -\sum_{k=1}^{K} \log \mathcal{L}(\widetilde{\pi}, \widetilde{\lambda})_k \cdot W_{dist} \cdot W_{num}$$

$$= -\sum_{k=1}^{K}\sum_{i=1}^{M}\sum_{j=1}^{M} \log \mathcal{L}(\widetilde{\pi}, \widetilde{\lambda})_{i,j,k} \cdot \frac{w_{dist_{i,j}}}{\sum_{i,j=1}^{M} w_{dist_{i,j}}} \cdot \frac{w_{num_{i,j}}}{\sum_{i,j=1}^{M} w_{num_{i,j}}} \tag{7}$$

## 4 EXPERIMENTS

We evaluate the task-agnostic GeoVeX vectors with a qualitative analysis of the latent structures using cosine similarities, and with a quantitative analysis of the improvement of performances on down-stream task.

### 4.1 QUALITATIVE EXPERIMENTS

In this analysis, we perform a qualitative comparison between GeoVeX and Hex2Vec embedding vectors, where both models are trained on the same exact worldwide input data from H3 and OSM

(200+ millions vectors worldwide), and with the same size of embeddings space (32), to not provide any advantage based on a different representation space. Space2Vec embeddings do not represent any geographical characteristic so they cannot be used for unsupervised geographical explorations.

First, we selected a sample of different cities all around the world, from different continents, each characterized by a mix of urban, suburban, green and water regions: Los Angeles, London, Cape Town, Tokyo and Mexico City. Then, we calculated the cosine similarity between an anchor point located in downtown (the exact center of the image) and each other point located in the region, to evaluate the potential discovery of clusters of locations with similar characteristics. The cosine similarities are shown in Fig. 5: the blue color indicates higher similarity to the center point location, and the red color indicates lower similarity.

The cosine similarity calculated using the GeoVex embeddings is shown in upper images: we can see that the geographic semantic captured by the model is aligned to human expectations, to some degree: adjacent hexagons tend to have similar embeddings until they are very different in terms of nearby geographical entities, and clusters of locations can be defined to delineate similar regions.

The cosine similarity calculated using Hex2Vec is shown in bottom images from Fig. 5: in this case, there is no evident human-readable semantic that can be extracted by this plot. We can conclude that the Hex2Vec model formulation does not perform well in terms of assigning similar embeddings to adjacent hexagons, as there are very few smooth clusters of hexagons that can be seen inside these more noisy patterns.

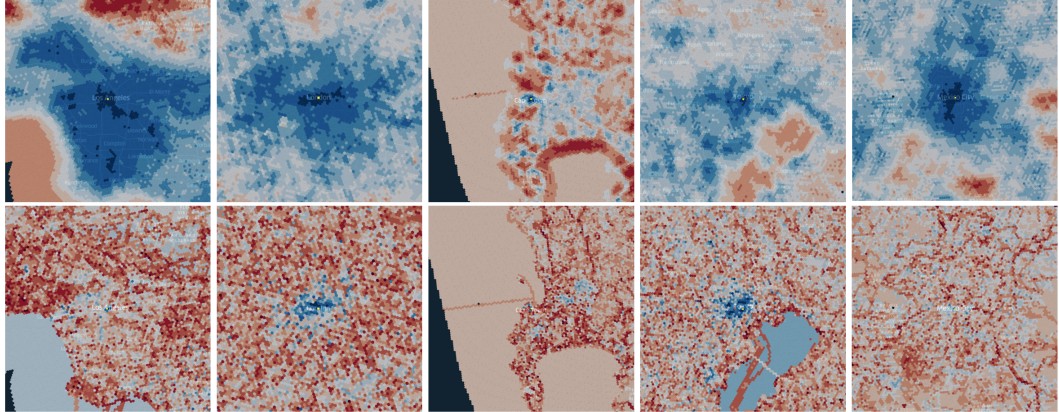

Figure 5: The cosine similarity calculated using embedding vectors from **GeoVeX (top)** and **Hex2Vec (bottom)**, using the yellow hexagon of the center point location as anchor item, for Los Angeles, London, Cape Town, Tokyo and Mexico City.

## 4.2 QUANTITATIVE EXPERIMENTS

We conducted quantitative experiments on the impact of using GeoVeX vectors as additional features in two downstream tasks: 1) prediction of worldwide vacation rental listings price in the travel industry (Sec. 4.2) and 2) hyperlocal interpolation of temperatures from weather stations (Sec. A.6 in Appendix). In both cases, the embeddings are pre-trained, agnostic of downstream tasks, frozen and then used as-is as additional features, to demonstrate that the task-agnostic learnt representations are meaningful.

Expressiveness of GeoVeX, Hex2Vec and Space2Vec vectors is evaluated with a **feature ablation study**, by comparing models using: 1) Common task features (Baseline); Additional features: 2) GeoVeX vectors; 3) Hex2Vec vectors; 4) Space2Vec encoding vectors; 5) both GeoVeX and Space2Vec; 6) both Hex2Vec and Space2Vec.

All competing models are trained with the same Gradient Boosting Machines (GBM) regression model architecture, which enables the discovery of interactions among features. For each task, the model hyperparameters are optimized through random search (see Sec. A.4 of Appendix for details on how to reproduce the experiments setup). For fair comparison, Space2Vec hyper-parameters

have been optimized for best task performance, since this model requires manual parameters to be configured, and Hex2Vec pre-trained embeddings are learnt with same data and same embedding size of GeoVeX.

**Vacation Rentals Price Prediction Task**  Open-source Inside Airbnb dataset[3] contains 1M+ listings located worldwide and describes each vacation rental listing with lodging characteristics, geographical coordinates, neighborhood name and daily price, which is the value to predict, after a conversion to a common currency. The hypothesis is that the geographical characteristics of the location influence its price, and thus GeoVeX vectors improve model performance when added as contextual geographical features. See Section A.5 of Appendix for the list of cities and regions obtained, so to replicate the experiment setup.

In terms of features, the baseline model is designed with the set of state-of-the-art features form (Islam et al., 2022), where base geographical information is composed of latitude, longitude and neighborhood name. Its intent is not be used as a performance benchmark, but to underline how much information the additional geospatial embeddings bring on top of a state-of-the-art set of task-specific features. The loss function is the Mean Squared Error (MSE) of the logarithm of the price, and the metrics monitored are both the loss and the Mean Absolute Error (MAE) of the price. The listings are split into training, validation and test with ratios 80%;10%;10%, with two splitting methods: a common random split, and a split by city names, to evaluate how well the model is able to generalize to new unseen geographical areas (i.e. out of training space). Table 1 shows the average performance of all models trained 7 times, together with the respective standard deviation.

We can see that the GeoVeX vectors are useful in both test strategies. For unseen cities, the GBM model with GeoVeX vectors confidently outperforms every other tested model, suggesting that GeoVeX improves both upon Hex2Vec, to compress geographical information useful to predict out-of-space, and upon Space2Vec, which encodes the location coordinates without considering the geographical entities. For unseen coordinates (random split), the GBM model with GeoVeX vectors paired with Space2Vec is the only model able to confidently improve the baseline performance.

Table 1: Experiment results for the price prediction task. The lower the better.

| Model | Split Randomly | | Split By City | |
|---|---|---|---|---|
| | MSE ln(Price) | MAE Price | MSE ln(Price) | MAE Price |
| Baseline | $0.281 \pm 0.005$ | $51.61 \pm 0.46$ | $0.492 \pm 0.017$ | $72.69 \pm 1.15$ |
| w/ GeoVeX | $0.285 \pm 0.008$ | $52.29 \pm 0.61$ | $\mathbf{0.465 \pm 0.007}$ | $\mathbf{70.30 \pm 0.26}$ |
| w/ Hex2Vec | $0.286 \pm 0.010$ | $52.20 \pm 0.91$ | $0.489 \pm 0.008$ | $71.33 \pm 0.42$ |
| w/ Space2Vec | $0.290 \pm 0.013$ | $52.51 \pm 1.17$ | $0.508 \pm 0.010$ | $72.56 \pm 0.61$ |
| w/ GeoVeX & Space2Vec | $\mathbf{0.271 \pm 0.009}$ | $\mathbf{50.65 \pm 0.89}$ | $0.492 \pm 0.010$ | $71.99 \pm 0.66$ |
| w/ Hex2Vec & Space2Vec | $0.280 \pm 0.004$ | $51.59 \pm 0.35$ | $0.487 \pm 0.010$ | $72.53 \pm 0.44$ |

## 5  CONCLUSION

We presented GeoVeX, a representation model which learns geospatial vectors as latent representations of the location's geographical characteristics, contextualized on the neighboring locations. GeoVeX is based on a novel architecture named Hexagonal Convolutional Autoencoder (HCAE) with a Zero-Inflated Poisson (ZIP) reconstruction layer to learn from hexagonal grids described with zero-inflated counts of the geographical tags of OSM entities. Qualitative analysis showcases the improvement of representation structures compared to state-of-the-art Hex2Vec model. Then, quantitative experiments on real open data support the fact that the pre-trained GeoVeX embeddings can improve models in multiple domains without task-specific re-training. In future, we aim to analyze the impact of changing the H3 grid resolution, test different zero-inflated probabilistic distributions, compare the task-agnostic vectors against some task-specialized versions, and tackle a broader set of downstream tasks that involve geolocated entities, such as: climate-related tasks (e.g. effect of urbanization on temperatures), classifications of geolocated images, or ranking of geolocated items.

---

[3]http://insideairbnb.com

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

# A    APPENDIX

## A.1    MATHEMATICAL FORMULA OF 3X3 MASKED HEXAGONAL CONVOLUTION

The mathematical formulation of the 3x3 masked hexagonal convolution is the following:

$$Y_k = H_k * X_k = (W_k \odot M_k) * X_k = (\begin{bmatrix} w_{-1,1} & w_{0,1} & w_{1,1} \\ w_{-1,0} & w_{0,0} & w_{1,0} \\ w_{-1,-1} & w_{0,-1} & w_{1,-1} \end{bmatrix} \odot \begin{bmatrix} 0 & 1 & 1 \\ 1 & 1 & 1 \\ 1 & 1 & 0 \end{bmatrix}) * X_k \quad (8)$$

where $H$ is the 3x3 masked hexagonal convolution kernel; $W$ is the standard 3x3 convolution kernel; $M$ is the 3x3 hexagonal mask; $X$ is the input tensor; $*$ is the convolution operation; $\odot$ is the element-wise multiplication operation; $i$ and $j$ are the two axial coordinates; $k$ is the 3rd channel of the tensor, representing the normalized count of the $k$-th OSM tag in the first convolution, and the $k$-th normalized filter in the subsequent convolutions.

## A.2    VISUAL REPRESENTATION OF THE LOSS KERNELS

In this section, we show the plots of the weighting kernels used in the loss formulation: the distance weighting kernel and numerosity weighting kernel. The kernels are plotted in the axial coordinates system, projected to a 2D matrix form. For each $i, j$ pair of coordinates, the weight is displayed.

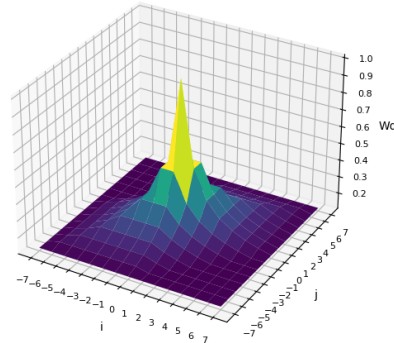

Figure 6: The distance weighting kernel.    Figure 7: The numerosity weighting kernel.

## A.3    H3 AND OPENSTREETMAP DATA PREPARATION

The construction of the database of hexagons described by respective geographical entities is done as follows. First, we calculate the set of hexagons that lie within the boundary of each country shape, using resolution 8 (best trade-off between data granularity, computational power and memory needed to compute the expensive geospatial operations). The result is a set of 200+ millions hexagons to cover the Earth's lands. Then, from the worldwide OSM database of geographical entities, we extract: 150 millions of *points*, identified by their coordinates; 220 millions of *lines*, identified by a set of coordinates; 500 millions of *multi-polygons*, consisting of one or more set of coordinates describing the polygons, and one or more set of coordinates describing the holes in the polygons. These three types of entities are geographically intersected with the hexagons, and a Bag-Of-Words (BOW) model is used to count the OSM geographical tags attributed to the entities intersecting each hexagon.

The extraction of the set of points, lines and polygons covering all the meaningful entities that can be used to describe the world is done with GDAL *ogr2ogr* [4]. The initial selection of OSM keys is done based on both the outcome of the analysis done in Woźniak & Szymański (2021), with some

---

[4]https://gdal.org/programs/ogr2ogr.html

other tags kept and cleaned since relevant to our extended use case (e.g. highways). In addition, we filtered out the tags with the value containing the following post-fixes: *:yes* and *:no*, since we found them to be not consistent and misleading. We also filtered out tags that are only available in few countries, so not relevant worldwide.

The geographical operations to clean, manipulate and intersect OSM entities with H3 hexagons are achieved using Apache Sedona [5] on an Apache Spark [6] cluster with a total of 10 Tb of memory and 1280 cores. Once the geospatial joins of hexagons and OSM entities are computed, a second level of sort and filtering is performed, in order to select the most meaningful features used to describe the hexagons of the world. In fact, based on the analysis of aggregated statistics, it is clear that some *key=value* pairs are only valid in few parts of the world (e.g. administrative tags), so we keep only pairs that are assigned to hexagons in at least 20 countries. Finally, we sort the pairs by descending distinct count of hexagons with that pair assigned, and we select the top 1024 tags. This cut-off is chosen using a trade-off between the quality of additional descriptors and the GPU memory constraint during model training.

## A.4 HYPERPARAMETER SELECTION FOR TASKS MODELS

For each task and for each model to train, we optimize the following LightGBM hyper-parameters, by performing a random search with 60 trials, since it can be proven that if the close-to-optimal region of hyperparameters occupies at least 5% of the grid surface, then random search with 60 trials will find that region with high probability (Bergstra & Bengio, 2012):

- *max_depth*: random int from 0 to 32;
- *num_leaves*: random int from 4 to 2048;
- *min_data_in_leaf*: random int from 20 to 10000;
- *min_gain_to_split*: random int from 0 to 15;
- *max_cat_threshold*: random int from 32 to 1024;
- *bagging_fraction*: random float from 0.2 to 0.95, with step of 0.1;
- *feature_fraction*: random float from 0.3 to 0.95, with step of 0.1;
- *learning_rate*: a range of float from 0.001 to 0.2;
- *lambda_l1*: random int from 0 to 100, with step of 5;
- *lambda_l2*: random int from 0 to 100, with step of 5;
- *extra_trees*: random true or false.

For each model using *Space2Vec* coordinates encoding, the following additional hyper-parameters are optimized, using a range of values suggested by the authors:

- *lambda_min*: random value in: 0.1, 1, 10, 100, 500;
- *lambda_min*: random value in: 1000, 10000, 40000;
- *num_frequencies*: random value in: 8, 16, 32, 64.

## A.5 VACATION RENTALS PRICE PREDICTION TASK - DETAILS

The cities and regions for which we obtained listing prices from by inside-airbnb open data are the following: Antwerp, Asheville, Athens, Austin, Bangkok, Barcelona, Beijing, Belize, Bergamo, Berlin, Bologna, Bordeaux, Boston, Bristol, Broward County, Brussels, Buenos Aires, Cambridge, Cape Town, Chicago, Clark County (NV), Columbus, Copenhagen, Crete, Denver, Dublin, Edinburgh, Euskadi, Florence, Gauteng, Geneva, Girona, Greater Manchester, Hawaii, Hong Kong, Istanbul, Jersey City, Lisbon, London, Los Angeles, Lyon, Madrid, Malaga, Mallorca, Menorca, Mexico City, Milan, Montreal, Munich, Naples, Nashville, New Brunswick, New Orleans, New York City, Oakland, Oslo, Ottawa, Pacific Grove, Paris, Portland, Porto, Prague, Puglia, Quebec City,

---

[5]https://sedona.apache.org/
[6]https://spark.apache.org/

Rhode Island, Rio De Janeiro, Rome, Salem (OR), San Diego, San Francisco, San Mateo County, Santa Clara County, Santa Cruz County, Santiago, Seattle, Sevilla, Shanghai, Sicily, Singapore, South Aegean, Stockholm, Taipei, Thessaloniki, Tokyo, Toronto, Trentino, Twin Cities (MSA), Valencia, Vancouver, Vaud, Venice, Victoria, Vienna, Washington DC, Zurich.

### A.6 HYPERLOCAL TEMPERATURES INTERPOLATION TASK

ARPA Lombardia is a regional agency for environment protection which provides open weather sensors data[7] in the area of Milan (Italy). The data contains 1.4 millions of hourly temperature samples, recorded by 13 weather stations from January 2010 to August 2022 in Milan and nearby. Each sample is described by: the temperature, the date, the hour, and the coordinates of the weather station. The task is the interpolation of the temperature between weather stations, which is modeled as a prediction of the temperature in an unknown location. This is achieved by removing a subset of stations from the training data and by comparing the model predictions with the actual temperatures observed by the test stations. The hypothesis is that the geographical characteristics of the location help to explain the differences in recorded temperatures from nearby sensors.

The experimental setup is the same as in Sec. 4.2. Baseline model denotes the model relying on all the available features when geospatial data are only latitude and longitude which are the date components (year, month and week of year) and the hour of the day. The loss function is the MSE of the temperature, and the metrics monitored are both the loss and the MAE. The temperature records are split into training, validation and test with ratios 80%;10%;10% by randomly sampling the sensors, to analyze the performance of estimating the temperatures at an unknown location.

Table 2 shows the performance of the different models on the test set. We can see that the GBM models with GeoVeX emebddings (both alone and combined to Space2Vec) confidently outperform every other model. This shows that GeoVeX vectors help in improving the estimation of temperatures in this region with a mix of urban and green areas, by providing the geographical context of each location.

Table 2: Experiment results for the hyperlocal temperature interpolation task. The lower the better.

| Model | MSE Temperature | MAE Temperature |
|---|---|---|
| Baseline | $11.144 \pm 0.094$ | $2.651 \pm 0.012$ |
| w/ GeoVeX | $10.656 \pm 0.135$ | $2.615 \pm 0.016$ |
| w/ Hex2Vec | $10.862 \pm 0.110$ | $2.628 \pm 0.010$ |
| w/ Space2Vec | $10.880 \pm 0.105$ | $2.639 \pm 0.012$ |
| w/ GeoVeX & Space2Vec | $\mathbf{10.651 \pm 0.090}$ | $\mathbf{2.613 \pm 0.012}$ |
| w/ Hex2Vec & Space2Vec | $10.870 \pm 0.172$ | $2.629 \pm 0.019$ |

---

[7]https://www.arpalombardia.it/Pages/Meteorologia/Richiesta-dati-misurati.aspx

