# OpenReview forum: "GeoVeX: Geospatial Vectors with Hexagonal Convolutional Autoencoders"
_ICLR.cc/2023/Conference — Submitted to ICLR 2023_

### Official Review · Reviewer_Fj26 · 2022-10-19

**Confidence:** 3
**Correctness:** 2
**Technical Novelty And Significance:** 3
**Empirical Novelty And Significance:** 2
**Recommendation:** 3

**Clarity, Quality, Novelty And Reproducibility:**

# Clarity
* The paper is generally well-written, with clear tables and figures.

# Novelty
* The proposed ZIP head for the decoder is novel to the best of my knowledge.
* Some of the other novelty claims made by the paper are questionable - the details are provided in the "Weaknesses" section above.

# Reproducibility
* There is not sufficient information in the paper to reproduce the results e.g. hyperparameter settings, tuning procedures. The paper makes reference to an appendix, but it does not seem to be included. Since this paper is meant to demonstrate the superiority of the proposed method, is important to provide details about how all methods (especially competing methods) were trained and tuned.
* How was the data split for the downstream tasks? Uniformly at random?
* What is the embedding dimensionality for GeoVeX, Space2Vec, and Hex2Vec? If the dimensions are different, does this provide unfair advantages to some methods?


**Strength And Weaknesses:**

# Strengths
* Learning geospatial representations that generalize to many downstream tasks is an interesting and worthwhile goal.
* The idea of autoencoding OSM data is a reasonable approach to geospatial representation learning.
* The proposed ZIP head is novel to the best of my knowledge.

# Weaknesses
* A key claim of the paper is that it produces "the first geospatial vectors trained on worldwide open data" - this does not seem to be true as written. For instance, [mac2019presence] learns embeddings for each location on earth from freely available data from iNaturalist. See also [sheehan2019predicting], which represents locations using embeddings of nearby geo-located Wikipedia articles.
* A key claim of the paper is that the proposed method learns "task-agnostic global geospatial vectors." Are they task agnostic? It is not obvious to me that using OSM data makes it "task agnostic" automatically, since OSM contains a limited set of categories with a particular focus on certain land use / land cover categories. More broadly, if the claim is that the proposed method learns significantly more general features than prior methods, it would probably make sense to demonstrate this on more than two datasets.
* A key claim of that paper is that it is "the first use of hexagonal convolutions within autoencoder architectures." First, I would like to be convinced that this is significant by itself. Can't hexagonal convolutions can be used with any convolutional architecture? Is there a reason we wouldn't expect it to work for autoencoders? Second, it seems like [hahner2022mesh] may have done this first - please comment and clarify.
* It's a bit tricky to interpret the results in Table 1 and Table 2 without more context. The magnitude of the performance differences between the methods seems small, and the ranking of methods seems to depend on the metric. Given this, it may be important to report average results across a few training runs (both the representation learning and the GBM) to determine whether we are observing robust performance differences or random fluctuations.
* GeoVeX several components, including the use of hexagonal spatial units, the ZIP head, and special terms in the loss function, none of which are characterized with an ablation study. The relative importance of these components is therefore unclear.

Minor comments:
* The paper claims that a linear probe does not work well in this context because GBM does better than the linear probe when combined with the baseline features. But isn't this what we'd expect if the baseline features are not very good? We would expect linear regression to work better with more powerful features.
* Why are GeoVeX features always combined with task-specific features (Figure 1)? Why not evaluate the utility of the features on their own for downstream tasks?
* In the related work section, the paper notes that Space2Vec "uses trigonometric functions with different frequencies to model a given position in space... However, the embeddings are not learnt based on the nearby geographical entities..." - note that [mac2019presence] does both of those things, where the geographic entities are species observations.

@inproceedings{sheehan2019predicting,
  title={Predicting economic development using geolocated wikipedia articles},
  author={Sheehan, Evan and Meng, Chenlin and Tan, Matthew and Uzkent, Burak and Jean, Neal and Burke, Marshall and Lobell, David and Ermon, Stefano},
  booktitle={Proceedings of the 25th ACM SIGKDD international conference on knowledge discovery \& data mining},
  pages={2698--2706},
  year={2019}
}

@inproceedings{mac2019presence,
  title={Presence-only geographical priors for fine-grained image classification},
  author={Mac Aodha, Oisin and Cole, Elijah and Perona, Pietro},
  booktitle={Proceedings of the IEEE/CVF International Conference on Computer Vision},
  pages={9596--9606},
  year={2019}
}

@inproceedings{hahner2022mesh,
  title={Mesh Convolutional Autoencoder for Semi-Regular Meshes of Different Sizes},
  author={Hahner, Sara and Garcke, Jochen},
  booktitle={Proceedings of the IEEE/CVF Winter Conference on Applications of Computer Vision},
  pages={885--894},
  year={2022}
}

**Summary Of The Paper:**

This paper aims to learn general-purpose location embeddings which are useful for downstream geospatial prediction tasks. The idea is to learn this representation by binning geographic entity occurrence data from OpenStreetMap (OSM) and applying a convolutional autoencoder with a special custom decoder head. The binning and convolution are done on a hexagonal grid. The proposed method (GeoVeX) is compared against two competing approaches to geospatial representation learning (Hex2Vec and Space2Vec) on two datasets.

**Summary Of The Review:**

This paper tackles an interesting problem and I enjoyed reading it. However, some of the (rather strong) claims made by the paper seem to be inadequately supported or incorrect. In addition, there are some important details missing (hyperparameters, tuning procedures, split construction) which makes it difficult to take the experimental results at face value.

---

> ### Author Response · Authors · 2022-11-18
> **Thank you very much for your valuable review and sharing these relevant references that we missed! We include and discuss these approaches in the new version of the paper. (1/4)**
>
> Thank you very much for your valuable review and sharing these relevant references that we missed! We include and discuss these approaches in the new version of the paper.
>
> Regarding your concerns:
>
>  > "W1: A key claim of the paper is that it produces "the first geospatial vectors trained on worldwide open data" - this does not seem to be true as written. For instance, [mac2019presence] learns embeddings for each location on earth from freely available data from iNaturalist. See also [sheehan2019predicting], which represents locations using embeddings of nearby geo-located Wikipedia articles."
>
>  Thank you for pointing out that the claim is not true as it was written. We have modified the statements regarding open data to explicitly focus on GIS databases (OpenStreetMap in our case) since they can describe all Earth locations, while other open dataset do not (they contain geo-located articles for a set of coordinates). In fact, as specified in the Problem Formulation (Sec 3.1) the objective is to learn pre-trained embeddings that can cover any location on Earth, without manual task-specific imputation for out-of-training-space coordinates, and without training them specifically for a task.
>
> Given this formulation, regarding the mentioned papers, in [mac2019presence] the location encoder is specifically learnt from photos to improve a set of image-classification tasks, so the pre-trained model represents only the embedding of the given set of photos locations, which cannot be used as-is in locations outside the given scope, unless we impute embeddings to new locations; in [sheehan2019predicting] the authors learn task-specific location embeddings from geolocated Wikipedia articles, which, again, cover a limited set of coordinates, thus requiring manual imputation logic for unknown locations.
>
> > "W2: A key claim of the paper is that the proposed method learns "task-agnostic global geospatial vectors." Are they task agnostic? It is not obvious to me that using OSM data makes it "task agnostic" automatically, since OSM contains a limited set of categories with a particular focus on certain land use / land cover categories. More broadly, if the claim is that the proposed method learns significantly more general features than prior methods, it would probably make sense to demonstrate this on more than two datasets."
>
>  We confirm that we pre-train the embeddings without any task-specific loss (as it is instead the case for [mac2019presence]  and  [sheehan2019predicting]), but only with a pure reconstruction loss for a zero-inflated counting process distributed on an hexagonal grid. So the training method is indeed task-agnostic. We did not filter on any task-specific OpenStreetMap tag (e.g. specific land use), since the aim is to have learnt representation geographically as complete as possible.
>
> >  "W3: A key claim of that paper is that it is "the first use of hexagonal convolutions within autoencoder architectures." First, I would like to be convinced that this is significant by itself. Can't hexagonal convolutions can be used with any convolutional architecture? Is there a reason we wouldn't expect it to work for autoencoders? Second, it seems like [hahner2022mesh] may have done this first - please comment and clarify."
>
>  Thank you for mentioning the paper [hahner2022mesh]! We have honestly missed this recent reference and we have added it in the Related Work section in the new submission!
>
>  We agree that the claim is not true as it was written, since [hahner2022mesh] proposed hexagonal convolutions within an autoencoder architecture in the context of 3D surfaces. In the new submission, we have rephrased to "it introduces a zero-inflated Poisson autoencoder reconstruction layer, to adapt a standard autoencoder network to train on sparse geographical count data distributed on an hexagonal grid". We believe this claim still holds true, since, to the best of our knowledge, GeoVeX model is the first one that adapted the hexagonal convolutional autoencoder network to work with count data instead of images/surfaces. This modification requires a change in how the decoder works (we propose a ZIP head) since otherwise the reconstruction process is not align to the known counting phenomenon that creates the data to reconstruct.

---

> > ### Author Response · Authors · 2022-11-18
> > **Thank you very much for your valuable review and sharing these relevant references that we missed! We include and discuss these approaches in the new version of the paper. (2/4)**
> >
> > > "W4: It's a bit tricky to interpret the results in Table 1 and Table 2 without more context. The magnitude of the performance differences between the methods seems small, and the ranking of methods seems to depend on the metric. Given this, it may be important to report average results across a few training runs (both the representation learning and the GBM) to determine whether we are observing robust performance differences or random fluctuations."
> >
> > In the new submission, we have run the experiments multiple times, to be more confident about our statements. Thanks for pointing this out, because by including the range of variability, and extending to worlwide regions instead of only US, GeoVeX appears as a winner in a more clear fashion!
> >
> >  >  "W5: GeoVeX several components, including the use of hexagonal spatial units, the ZIP head, and special terms in the loss function, none of which are characterized with an ablation study. The relative importance of these components is therefore unclear."
> >
> >  We know that the proposed model uses many new concepts if compared to the image domain, but the design of the architecture simply follows the type of data we are dealing with (geographical count data distributed on an hexagonal grid), so we did not do specific ablation study for elements that are either defined "by construction" or are shown to be better by previous works. However, thanks to your feedback we have improved clarity since we have added more references in the new submission!
> >
> >          - Hexagonal grid:
> >
> >               - A squared grid on geographical maps introduces distortions regarding the neighboring relations, so the hexagonal grid is the state-of-the-art representation of geographical grids, because it guarantees the isotropy of local neighbourhoods. See [wang2020isotropic] for more details.
> >
> >          - Zero-Inflated-Poisson (ZIP) head:
> >
> >                - Given that the underlying process is a counting process by construction, the Poisson distribution naturally fits the known counting phenomenon. Regarding "zero-inflation", we have referenced the test to perform to verify the "zero-inflation hypothesis" of the Poisson distribution, for each OSM count variable: [van1995score] and [yang2010score]. We performed the test, and it rejects the null hypothesis that the observed 0s follow the Poisson distribution with p-values at minimum of 9.8% and in >90% of cases with p-value of 100%, thus stating that the observed 0s of the Poisson distribution are inflated.
> >
> >         - Loss function:
> >
> >               - The loss function is naturally derived from the log-loss of the ZIP formulation, and from the design of the hexagonal grid: by using an hexagonal grid, the final loss must be weighted based on the grid design (each cell has 6 neighbors and each hexagonal ring has 6 more hexagons than the previous ring). The only component we chose is the form of the distance weight decay (the "inverse" function 1/x): we indeed use the simplest form of weight decay formulation to not add any ad-hoc decay function which would then require, as you correctly pointed out, an ablation study.
> >
> >          - Hexagonal convolutions:
> >
> >                - Hexagonal convolutions give better results than standard convolutions in case of hexagonal shaped data, as demonstrated by [hoogeboom2018hexaconv] and [steppa2019hexagdly], so we have not added any ablation study regarding this natural choice. We have added these references in the new submitted paper.
> >
> >           - Embedding size (bottleneck layer):
> >
> >               - The embedding size for GeoVeX and Hex2Vec is equal, in order to not provide any advantage based on a potential bigger representation space. We have added this information in the new version of the paper to be more clear. We plan to do a sensitivity analysis of this parameter in the future works: empirically we used the biggest size after which we did not notice any change in the learnt structures nor reconstruction loss.
> >
> >              -  For example, to be fair, the mentioned paper [hahner2022mesh] uses the same design choices regarding hexagonal grids, hexagonal convolutions, and fixed latent space size (8) without demonstration nor ablation.

---

> > > ### Author Response · Authors · 2022-11-18
> > > **Thank you very much for your valuable review and sharing these relevant references that we missed! We include and discuss these approaches in the new version of the paper. (3/4)**
> > >
> > > For the minor points:
> > >
> > >  > "M1: The paper claims that a linear probe does not work well in this context because GBM does better than the linear probe when combined with the baseline features. But isn't this what we'd expect if the baseline features are not very good? We would expect linear regression to work better with more powerful features."
> > >
> > >  You are right in saying that we expect that GBM works better than a linear model. The goal was just to show that the implementation of the GBM automated optimization procedure (60-iteration random search for every model setup) was implemented correctly, and not customized to be better or worse than a simple linear baseline. Since it seems misleading, we have removed the Linear baseline since it is there just for checking the implementation quality, and not to be actually compared to the other methods.
> > >
> > >  What we want to demonstrate, using a feature ablation study, is that, with same model architecture able to leverage features interaction (a GBM model), the addition of task-agnostic vectors can have impacts on the final performance, even if not learnt for a specific task in mind.
> > >
> > > > "M2: Why are GeoVeX features always combined with task-specific features (Figure 1)? Why not evaluate the utility of the features on their own for downstream tasks?"
> > >
> > > Real-world tabular tasks come with a set of base features that are very predictive (e.g. the type of property to book for a price prediction task), so we want to demonstrate that adding extra geographic information not available in the raw task dataset can improve task model performance, even if not learnt with that specific task in mind. We did not want to focus on solving real-world tasks by ignoring information we actually know, achieving at the end sub-optimal performance, but instead we wanted to start from solid baselines.
> > >
> > > >  "M3: In the related work section, the paper notes that Space2Vec "uses trigonometric functions with different frequencies to model a given position in space... However, the embeddings are not learnt based on the nearby geographical entities..." - note that [mac2019presence] does both of those things, where the geographic entities are species observations."
> > >
> > > Thank you very much for pointing this out! In the comment for W1, also now reported in the new version of the paper (Sec. 2), we discuss the differences between this approach and ours.
> > >
> > > Regarding reproducibility:
> > >
> > > >  "R1: There is not sufficient information in the paper to reproduce the results e.g. hyperparameter settings, tuning procedures. The paper makes reference to an appendix, but it does not seem to be included. Since this paper is meant to demonstrate the superiority of the proposed method, is important to provide details about how all methods (especially competing methods) were trained and tuned."
> > >
> > > Thank you for your feedback! As per conference requirements, we included the appendix in the "Supplementary Material" section, and not in the main document. However, since we have seen that reviewers may have difficulties in connecting appendix with main paper, the new submission has the appendix included in the main document. We are open to change the submission format to stick with the rules.
> > >
> > > >  "R2: How was the data split for the downstream tasks? Uniformly at random?"
> > >
> > > For both tasks, the main split was done randomly, and for the specific price-prediction task, we added a custom split by city name, so that the model was also tested in case of predicting unseen geographic regions outside the training space. In this scenario, the geographical embeddings express their best potential since they enable more "transfer learning" to new geographic regions, without any manual imputation based on some nearest-embeddings rules.
> > >
> > > >  "R3: What is the embedding dimensionality for GeoVeX, Space2Vec, and Hex2Vec? If the dimensions are different, does this provide unfair advantages to some methods?"
> > >
> > >  The embedding size for GeoVeX and Hex2Vec is equal, in order to not provide any advantage based on a potential bigger representation space. We have added this information in the new version of the paper to be more clear. We plan to do a sensitivity analysis of this parameter in the future works: empirically we used the biggest size after which we did not notice any change in the learnt structures nor reconstruction loss.

---

> > > > ### Author Response · Authors · 2022-11-18
> > > > **Thank you very much for your valuable review and sharing these relevant references that we missed! We include and discuss these approaches in the new version of the paper. (4/4)**
> > > >
> > > > [wang2020isotropic] Wang, Lu, et al. "The isotropic organization of DEM structure and extraction of valley lines using hexagonal grid." Transactions in GIS 24.2 (2020): 483-507.
> > > >
> > > > [van1995score] Van den Broek, Jan. "A score test for zero inflation in a Poisson distribution." Biometrics (1995): 738-743.
> > > >
> > > > [yang2010score] Yang, Zhao, James W. Hardin, and Cheryl L. Addy. "Score tests for zero-inflation in overdispersed count data." Communications in Statistics—Theory and Methods 39.11 (2010): 2008-2030.
> > > >
> > > > [hoogeboom2018hexaconv] Hoogeboom, Emiel, et al. "Hexaconv." arXiv preprint arXiv:1803.02108 (2018).
> > > >
> > > > [steppa2019hexagdly] Steppa, Constantin, and Tim L. Holch. "HexagDLy—Processing hexagonally sampled data with CNNs in PyTorch." SoftwareX 9 (2019): 193-198.

---

> > > > > ### Comment · Reviewer_Fj26 · 2022-12-06
> > > > > **Thank you for the response!**
> > > > >
> > > > > Thank you for the response. I will focus on outstanding concerns below.
> > > > >
> > > > > > We have modified the statements regarding open data to explicitly focus on GIS databases (OpenStreetMap in our case) since they can describe all Earth locations, while other open dataset do not (they contain geo-located articles for a set of coordinates). In fact, as specified in the Problem Formulation (Sec 3.1) the objective is to learn pre-trained embeddings that can cover any location on Earth, without manual task-specific imputation for out-of-training-space coordinates, and without training them specifically for a task.
> > > > >
> > > > > Thank you for updating the claim, but I still disagree with these statements. I don't think it's fair to say that OpenStreetMap can "describe all Earth locations" while e.g. Wikipedia (as used in [sheehan2019predicting]) or iNaturalist (as used in [mac2019presence]) cannot. Both are simply collections of volunteered information, with their own peculiar biases and incomplete coverage. Is there a reference that supports a claim like: "OpenStreetMap has high-quality data for every country on earth"? I suspect that the high-quality data is extremely skewed towards e.g. the U.S. and Europe (as is also true for Wikipedia and iNaturalist).
> > > > >
> > > > > > Given this formulation, regarding the mentioned papers, in [mac2019presence] the location encoder is specifically learnt from photos to improve a set of image-classification tasks, so the pre-trained model represents only the embedding of the given set of photos locations, which cannot be used as-is in locations outside the given scope, unless we impute embeddings to new locations; in [sheehan2019predicting] the authors learn task-specific location embeddings from geolocated Wikipedia articles, which, again, cover a limited set of coordinates, thus requiring manual imputation logic for unknown locations.
> > > > >
> > > > > I believe these statements are incorrect. [sheehan2019predicting] can predict at any location (whether in the training set or not) using night-time lights and the Wikipedia articles within a certain radius of a query location. [mac2019presence] learns their location embeddings from arbitrary species occurrence data, which does not have to be paired with images. Both [sheehan2019predicting] and [mac2019presence] can produce embeddings of arbitrary locations and there is no "manual imputation logic" as far as I know. While those embeddings may be "good" or "bad" depending on the location, that's a separate question concerning generalization.
> > > > >
> > > > > > We confirm that we pre-train the embeddings without any task-specific loss (as it is instead the case for [mac2019presence] and [sheehan2019predicting]), but only with a pure reconstruction loss for a zero-inflated counting process distributed on an hexagonal grid. So the training method is indeed task-agnostic. We did not filter on any task-specific OpenStreetMap tag (e.g. specific land use), since the aim is to have learnt representation geographically as complete as possible.
> > > > >
> > > > > The loss may be task-agnostic, but that doesn't make the features task agnostic as claimed ("task-agnostic global geospatial vectors"). If we train a convolutional autoencoder on chest X-rays, we would not say the resulting features are task-agnostic. Similarly, OSM is a particular dataset with its own strengths and limitations - I don't see any reason to believe that autoencoding OSM data leads to "task agnostic global geospatial vectors" automatically. It is possible that these features will turn out to be much more broadly useful than e.g. [mac2019presence] and [sheehan2019predicting], but that remains to be shown. Since GeoVeX was tested on only two downstream tasks, broad task generality was not shown.
> > > > >
> > > > > > In the new submission, we have run the experiments multiple times, to be more confident about our statements. Thanks for pointing this out, because by including the range of variability, and extending to worlwide regions instead of only US, GeoVeX appears as a winner in a more clear fashion!
> > > > >
> > > > > Thank you for this addition. See comments below.
> > > > >
> > > > > > We know that the proposed model uses many new concepts if compared to the image domain, but the design of the architecture simply follows the type of data we are dealing with (geographical count data distributed on an hexagonal grid), so we did not do specific ablation study for elements that are either defined "by construction" or are shown to be better by previous works. However, thanks to your feedback we have improved clarity since we have added more references in the new submission!
> > > > >
> > > > > I think these ablations are important, especially because the performance margins are razor thin between the proposed method and prior work. In fact, Hex2Vec often outperforms GeoVeX in direct comparisons. The comparisons where GeoVeX wins are usually in combination with Space2Vec - but this suggests that GeoVeX and Space2Vec are more complementary than Hex2Vec and Space2Vec, not that GeoVeX is itself better than Hex2Vec.

---

> > > > > > ### Author Response · Authors · 2022-12-07
> > > > > > **Thank you very much for reviewing again the paper! (1/2)**
> > > > > >
> > > > > > Thank you very much for reviewing again the paper!
> > > > > >
> > > > > > > Thank you for updating the claim, but I still disagree with these statements. I don't think it's fair to say that OpenStreetMap can "describe all Earth locations" while e.g. Wikipedia (as used in [sheehan2019predicting]) or iNaturalist (as used in [mac2019presence]) cannot. Both are simply collections of volunteered information, with their own peculiar biases and incomplete coverage. Is there a reference that supports a claim like: "OpenStreetMap has high-quality data for every country on earth"? I suspect that the high-quality data is extremely skewed towards e.g. the U.S. and Europe (as is also true for Wikipedia and iNaturalist).
> > > > > >
> > > > > > We passed through the paper and the comments again and we did not find where we claimed that :"OpenStreetMap has high-quality data for every country on earth". What we claim is that OSM is a GIS database so it is intended to describe the geography of a location on earth, while geolocated Wikipedia articles only represent geographical concepts with facts associated with a location. It also does not contain any polygon geographies, while GIS databases contain them. So we argue that Wikipedia articles only represents partial geographical informations compared to GIS databases, otherwise we would not even need GIS databases in the first place.
> > > > > >
> > > > > > > I believe these statements are incorrect. [sheehan2019predicting] can predict at any location (whether in the training set or not) using night-time lights and the Wikipedia articles within a certain radius of a query location. [mac2019presence] learns their location embeddings from arbitrary species occurrence data, which does not have to be paired with images. Both [sheehan2019predicting] and [mac2019presence] can produce embeddings of arbitrary locations and there is no "manual imputation logic" as far as I know. While those embeddings may be "good" or "bad" depending on the location, that's a separate question concerning generalization.
> > > > > >
> > > > > > For "manual imputation logic", we intend that in [sheehan2019predicting] the authors use "10" nearby articles to train/predict in any location ("we averaged the embeddings of the 10 closest articles to it"), which is an hard-coded number with no ablation study (which may be different from task to task). With GeoVeX we want to improve this by not having to impute this number since we learn one specific embedding for each location, without averaging embeddings of a set of locations nearby or far away by kilometres ("that majority of the articles are within 100 km").
> > > > > >
> > > > > > In [mac2019presence] the locations covered are only the ones where the birds occur ("Our train locations and photographers are sampled from eBird"): we argue that this bird dataset does not describe the geography of a location (what is there in terms of buildings, roads, facilities, etc), but only the presence/absence of birds and species.
> > > > > >
> > > > > > > The loss may be task-agnostic, but that doesn't make the features task agnostic as claimed ("task-agnostic global geospatial vectors"). If we train a convolutional autoencoder on chest X-rays, we would not say the resulting features are task-agnostic. Similarly, OSM is a particular dataset with its own strengths and limitations - I don't see any reason to believe that autoencoding OSM data leads to "task agnostic global geospatial vectors" automatically. It is possible that these features will turn out to be much more broadly useful than e.g. [mac2019presence] and [sheehan2019predicting], but that remains to be shown. Since GeoVeX was tested on only two downstream tasks, broad task generality was not shown.
> > > > > >
> > > > > > We agree with you that the "task-agnostic" term is strong and is subject to interpretation. We will try in future to use a smoother term. We still believe that training GeoVeX with OSM (with its entities with a given a-priori semantic, as you correctly commented), is similar to training Word2Vec models with Wikipedia (biased to its set of articles, as OSM with its entities), which we consider task-agnostic because they can be used to augment the word semantics in text-related tasks without any parameter (a simple lookup of the embeddings). This is the idea we want to convey in the paper. We are open to suggestions to replace the term "task-agnostic".

---

> > > > > > > ### Author Response · Authors · 2022-12-07
> > > > > > > **Thank you very much for reviewing again the paper! (2/2)**
> > > > > > >
> > > > > > > > I think these ablations are important, especially because the performance margins are razor thin between the proposed method and prior work. In fact, Hex2Vec often outperforms GeoVeX in direct comparisons. The comparisons where GeoVeX wins are usually in combination with Space2Vec - but this suggests that GeoVeX and Space2Vec are more complementary than Hex2Vec and Space2Vec, not that GeoVeX is itself better than Hex2Vec.
> > > > > > >
> > > > > > > Thank you for your feedback. We are not sure what supports "Hex2Vec often outperforms GeoVeX" in the paper: the MSE of Hex2Vec models are always higher than the ones of GeoVeX models, both alone and when combined with Space2Vec.
> > > > > > > We agree with you, instead, that we show that in some tasks setups the GeoVeX and the Space2Vec models are complementary, so we wanted to demonstrate that even if a model already uses a geo-prior like Space2Vec, it can still benefit from GeoVeX vectors, since the combined model outperforms the Space2Vec model.

---

> > > > > > > > ### Comment · Reviewer_Fj26 · 2022-12-07
> > > > > > > > **Thanks for the engagement!**
> > > > > > > >
> > > > > > > > > We passed through the paper and the comments again and we did not find where we claimed that :"OpenStreetMap has high-quality data for every country on earth". What we claim is that OSM is a GIS database so it is intended to describe the geography of a location on earth, while geolocated Wikipedia articles only represent geographical concepts with facts associated with a location. It also does not contain any polygon geographies, while GIS databases contain them. So we argue that Wikipedia articles only represents partial geographical informations compared to GIS databases, otherwise we would not even need GIS databases in the first place.
> > > > > > > >
> > > > > > > > Apologies, I did not mean to suggest that this work has explicitly claimed that "OpenStreetMap has high-quality data for every country on earth". Rather, there seemed to be a claim that OpenStreetMap is a superior data source compared to e.g. geo-located Wikipedia articles, and I wanted to understand the basis for that claim. Based on the latest response, it sounds like the basis for that claim is that OpenStreetMap is a GIS database that can include polygon objects in addition to point objects. This is true, but that does not relate to data quality. OpenStreetMap is a collection of volunteered information with its own peculiar biases and incomplete coverage, just like e.g. Wikipedia or iNaturalist. While I believe that OpenSteetMap, Wikipedia, iNaturalist, and other volunteer-driven projects are incredibly useful, I think it is important to be clear about their limitations. In my view, the paper should either (a) present OpenStreetMap as a source of incomplete and biased data that is not intrinsically superior to other volunteered data or (b) demonstrate through references or experiments that OpenStreetMap is superior in concrete ways.
> > > > > > > >
> > > > > > > > > For "manual imputation logic", we intend that in [sheehan2019predicting] the authors use "10" nearby articles to train/predict in any location ("we averaged the embeddings of the 10 closest articles to it"), which is an hard-coded number with no ablation study (which may be different from task to task). With GeoVeX we want to improve this by not having to impute this number since we learn one specific embedding for each location, without averaging embeddings of a set of locations nearby or far away by kilometres ("that majority of the articles are within 100 km").
> > > > > > > >
> > > > > > > > The paper under consideration includes the following line: "Each hexagon’s neighborhood is defined by the $r$-rings neighboring cells as described in Fig. 2 for $r = 7$ rings around the point (0, 0) ($r = 7$ has been experimentally defined to fit our GPU memory constraints)." This seems similar to the "hard-coded number" that is referred to in [sheehan2019predicting], but just at training time vs. testing time. It is not clear that this represents a fundamental difference between the current paper and [sheehan2019predicting]. And there seems to be no such limitation for [mac2019presence].
> > > > > > > >
> > > > > > > > > In [mac2019presence] the locations covered are only the ones where the birds occur ("Our train locations and photographers are sampled from eBird"): we argue that this bird dataset does not describe the geography of a location (what is there in terms of buildings, roads, facilities, etc), but only the presence/absence of birds and species.
> > > > > > > >
> > > > > > > > It is true that [mac2019presence] trains at locations where they have geospatial data. (Presumably this is true for the paper under consideration as well, unless OpenStreetMap has complete data for every location?) This is sometimes bird observations (e.g. NABirds), sometimes other animals and plants (e.g. iNat), and sometimes general object categories (e.g. YFCC). But this does not prevent their method from generating embeddings at arbitrary locations, and there seems to be no "manual imputation logic" as far as I can see. It certainly may be that it's better to learn embeddings from buildings, roads, facilities, etc. (this probably depends on the downstream task), but that is not obvious a priori and would need to be shown.
> > > > > > > >
> > > > > > > > > We agree with you that the "task-agnostic" term is strong and is subject to interpretation. We will try in future to use a smoother term. We still believe that training GeoVeX with OSM (with its entities with a given a-priori semantic, as you correctly commented), is similar to training Word2Vec models with Wikipedia (biased to its set of articles, as OSM with its entities), which we consider task-agnostic because they can be used to augment the word semantics in text-related tasks without any parameter (a simple lookup of the embeddings). This is the idea we want to convey in the paper. We are open to suggestions to replace the term "task-agnostic".
> > > > > > > >
> > > > > > > > I understand better now, thank you. Would it then be true that [sheehan2019predicting] and [mac2019presence] are "task agnostic" in the same sense, since both can be used to generate embedding vectors for any location?

---

> > > > > > > > ### Comment · Reviewer_Fj26 · 2022-12-07
> > > > > > > > **Thanks for the engagement! (Part 2)**
> > > > > > > >
> > > > > > > > > Thank you for your feedback. We are not sure what supports "Hex2Vec often outperforms GeoVeX" in the paper: the MSE of Hex2Vec models are always higher than the ones of GeoVeX models, both alone and when combined with Space2Vec. We agree with you, instead, that we show that in some tasks setups the GeoVeX and the Space2Vec models are complementary, so we wanted to demonstrate that even if a model already uses a geo-prior like Space2Vec, it can still benefit from GeoVeX vectors, since the combined model outperforms the Space2Vec model.
> > > > > > > >
> > > > > > > > Apologies, I misread one of the tables. What seems to be true is that GeoVeX beats Hex2Vec on price prediction when splitting by city, but GeoVeX is not the clear winner when splitting randomly for price prediction or for the temperature prediction task (due to small margins compared to the std, or to disagreement between MSE and MAE). Given the number of moving parts in this method, ablation studies would go a long way towards demonstrating that these performance gains are robust and providing the community with an understanding of where they come from.

---

### Official Review · Reviewer_pWwW · 2022-10-24

**Confidence:** 3
**Correctness:** 3
**Technical Novelty And Significance:** 3
**Empirical Novelty And Significance:** 3
**Recommendation:** 6

**Clarity, Quality, Novelty And Reproducibility:**

Please refer to the strengths and weaknesses section. Overall, the paper is well written and novel.

**Strength And Weaknesses:**

-- Strengths:
- Approach is novel, technically sound, and relevant to many applications
- Great potential impact
- Article is well written and set in context

-- Weaknesses:
- Not enough evaluation was done to fully test the power of the learnt representations. Only two downstream tasks were tested without comparing performance to other strong approaches
- Further ablation studies are needed to understand better the proposed approach. How important is the use of the ZIP output layer? What is the performance without it? How GeoVex would perform using other geographic systems?

**Summary Of The Paper:**

Authors proposed GeoVEX as a framework for global representation learning in gespatial settings. The apprach leverages the H3 geospatial indexing system and data from Open Street Maps to create an embedding for each location on earth. An autoencoder like network architecture was introduced for the embedding generation which includes a novel hexagonal convolutional operation and adds a Zero-Inflated Poisson probabilistic output layer. The learnt representations were tested for the tasks of vacation rentals price prediction and temperature interpolation.

**Summary Of The Review:**

There is enough contribution for the paper to be accepted but it would be stronger with a better set of downstream task evaluations and comparisons to other approaches for those.

---

> ### Author Response · Authors · 2022-11-18
> **Thank you very much for your valuable feedback and your encouraging review! We have made some changes and we have submitted a new version of the paper!**
>
> Thank you very much for your valuable feedback and your encouraging review! We have made some changes and we have submitted a new version of the paper!
>
> > "W1: Not enough evaluation was done to fully test the power of the learnt representations. Only two downstream tasks were tested without comparing performance to other strong approaches."
>
>   -  The new submission should even better convey the improvement we are seeing during the experiments, since we extended the experiments data to worldwide regions, and we added a qualitative analysis of the geographic structures learnt by GeoVeX compared to Hex2Vec! Please refer to Section 4 of the new version of the paper!
>
> > "W2: Further ablation studies are needed to understand better the proposed approach. How important is the use of the ZIP output layer? What is the performance without it? How GeoVex would perform using other geographic systems?"
>
>   -  We know that the proposed model uses many new concepts if compared to the image domain, but the design of the architecture simply follows the type of data we are dealing with (geographical count data distributed on an hexagonal grid), so we did not do specific ablation study for elements that are either defined "by construction" or are shown to be better by previous works. However, thanks to your feedback we have improved clarity since we have added more references in the new submission!
>      -   Hexagonal grid:
>          -   A squared grid on geographical maps introduces distortions regarding the neighboring relations, so the hexagonal grid is the state-of-the-art representation of geographical grids, because it guarantees the isotropy of local neighbourhoods. See [wang2020isotropic] for more details.
>      -   Zero-Inflated-Poisson (ZIP) head:
>          -   Given that the underlying process is a counting process by construction, the Poisson distribution naturally fits the known counting phenomenon. Regarding "zero-inflation", we have referenced the test to perform to verify the "zero-inflation hypothesis" of the Poisson distribution, for each OSM count variable: [van1995score] and [yang2010score]. We performed the test, and it rejects the null hypothesis that the observed 0s follow the Poisson distribution with p-values at minimum of 9.8% and in >90% of cases with p-value of 100%, thus stating that the observed 0s of the Poisson distribution are inflated.
>       -  Loss function:
>           -  The loss function is naturally derived from the log-loss of the ZIP formulation, and from the design of the hexagonal grid: by using an hexagonal grid, the final loss must be weighted based on the grid design (each cell has 6 neighbors and each hexagonal ring has 6 more hexagons than the previous ring). The only component we chose is the form of the distance weight decay (the "inverse" function 1/x): we indeed use the simplest form of weight decay formulation to not add any ad-hoc decay function which would then require, as you correctly pointed out, an ablation study.
>       -  Hexagonal convolutions:
>          -   Hexagonal convolutions give better results than standard convolutions in case of hexagonal shaped data, as demonstrated by [hoogeboom2018hexaconv] and [steppa2019hexagdly], so we have not added any ablation study regarding this natural choice. We have added these references in the new submitted paper.
>      -   Embedding size (bottleneck layer):
>           -  The embedding size for GeoVeX and Hex2Vec is equal, in order to not provide any advantage based on a potential bigger representation space. We have added this information in the new version of the paper to be more clear. We plan to do a sensitivity analysis of this parameter in the future works: empirically we used the biggest size after which we did not notice any change in the learnt structures nor reconstruction loss.
>
> [wang2020isotropic] Wang, Lu, et al. "The isotropic organization of DEM structure and extraction of valley lines using hexagonal grid." Transactions in GIS 24.2 (2020): 483-507.
>
> [van1995score] Van den Broek, Jan. "A score test for zero inflation in a Poisson distribution." Biometrics (1995): 738-743.
>
> [yang2010score] Yang, Zhao, James W. Hardin, and Cheryl L. Addy. "Score tests for zero-inflation in overdispersed count data." Communications in Statistics—Theory and Methods 39.11 (2010): 2008-2030.
>
> [hoogeboom2018hexaconv] Hoogeboom, Emiel, et al. "Hexaconv." arXiv preprint arXiv:1803.02108 (2018).
>
> [steppa2019hexagdly] Steppa, Constantin, and Tim L. Holch. "HexagDLy—Processing hexagonally sampled data with CNNs in PyTorch." SoftwareX 9 (2019): 193-198.

---

### Official Review · Reviewer_zeEd · 2022-10-27

**Confidence:** 4
**Clarity, Quality, Novelty And Reproducibility:** 3. Clarity, Quality, Novelty And Repr…
**Correctness:** 3
**Technical Novelty And Significance:** 3
**Empirical Novelty And Significance:** 3
**Recommendation:** 5

**Strength And Weaknesses:**

Strength
- The paper is clearly written and experiments demonstrated GeoVeX's improvement.

Weakness
- The novelty of the paper is limited. The general idea (using OSM + H3) is very similar to the Hex2Vec paper.
- The paper lacks ablation study on understanding why the proposed method work.


**Summary Of The Paper:**

The author proposed to a new geo-encoding method called GeoVeX. The idea is to train an autoencoder on the OSM data projected to H3. Experiments show that GeoVeX outperforms baselines like Hex2Vec and Space2Vec.


**Summary Of The Review:**

Vote for weak reject due to lack of novelty.

---

> ### Author Response · Authors · 2022-11-18
> **Thank you very much for your valuable review! We have made some changes and we have submitted a new version of the paper! (part 1/2)**
>
> Thank you very much for your valuable review! We have made some changes and we have submitted a new version of the paper!
>
> > "S1: The paper is clearly written and experiments demonstrated GeoVeX's improvement."
>
> Thank you for your encouraging message! The new submission should even better convey the improvement we are seeing during the experiments, since we extended the experiments data to worldwide regions, and we added a qualitative analysis of the geographic structures learnt by GeoVeX compared to Hex2Vec!
>
> > "W1: The novelty of the paper is limited. The general idea (using OSM + H3) is very similar to the Hex2Vec paper."
>
> - Despite the similarities, there is fundamental difference between GeoVeX and Hex2Vec: Hex2Vec does not consider any information from the neighboring hexagons. As a consequence, Hex2Vec embeddings cannot be considered contextualized in the sense the learnt representation of an hexagon does not change based on where this hexagon is in the world. This can be problematic in terms of semantics as shown in the qualitative comparison between GeoVeX and Hex2Vec embedding vectors for the city of Los Angeles (paragraph A.1 in Appendix of the initial submission). While for GeoVeX, we can clearly recover the coast, city center, etc., there is nothing human-readable for the Hex2Vec vectors since each hexagon's embedding is not influenced by the surrounding hexagons.
>
> - Based on your relevant observations, we understood that the difference was not easy to understand, so we improved the Related Work section and we added a qualitative analysis (Section 4.1) where we integrated more analyses on the semantic captured by GeoVeX vectors and by Hex2Vec vectors. We can clearly see some geographic structures captured by GeoVeX, while we get a lot of noisy latent vectors in Hex2Vec, since not contextualized. You can find this analysis in the new version of the paper!

---

> > ### Author Response · Authors · 2022-11-18
> > **Thank you very much for your valuable review! We have made some changes and we have submitted a new version of the paper! (part 2/2)**
> >
> > > "W2: The paper lacks ablation study on understanding why the proposed method work."
> >
> > -    We perform a feature ablation study in terms of comparing GeoVeX vectors with both a baseline model and the state-of-the-art models in the experimentation section:
> >
> >         - Compared to the vectors from a state-of-the-art H3/OSM network architecture (Hex2Vec), we demonstrate improvements both qualitatively (Section 4.1, added in the new version of the paper) and quantitatively (Section 4.2, extended to worldwide regions).
> >
> >        - Compared to the vectors from a state-of-the-art geo prior (Space2Vec), which is agnostic of any geographical characteristic of the location, we demonstrate that the usage of embeddings pre-trained on an open geographical database (so not only a trigonometric reformulation of latitude and longitude) can be used to improve downstream tasks, expecially when predicting out-of-space (in cities not seen during training).
> >
> >
> > -    Regarding the ablation of network components, we know that the proposed model uses many new concepts if compared to the image domain, but the design of the architecture simply follows the type of data we are dealing with (geographical count data distributed on an hexagonal grid), so we did not do specific ablation study for elements that are either defined "by construction" or are shown to be better by previous works. However, thanks to your feedback we have improved clarity since we have added more references in the new submission!
> >       -  Hexagonal grid:
> >            - A squared grid on geographical maps introduces distortions regarding the neighboring relations, so the hexagonal grid is the state-of-the-art representation of geographical grids, because it guarantees the isotropy of local neighbourhoods. See [wang2020isotropic] for more details.
> >       -  Zero-Inflated-Poisson (ZIP) head:
> >            - Given that the underlying process is a counting process by construction, the Poisson distribution naturally fits the known counting phenomenon. Regarding "zero-inflation", we have referenced the test to perform to verify the "zero-inflation hypothesis" of the Poisson distribution, for each OSM count variable: [van1995score] and [yang2010score]. We performed the test, and it rejects the null hypothesis that the observed 0s follow the Poisson distribution with p-values at minimum of 9.8% and in >90% of cases with p-value of 100%, thus stating that the observed 0s of the Poisson distribution are inflated.
> >       -  Loss function:
> >           -  The loss function is naturally derived from the log-loss of the ZIP formulation, and from the design of the hexagonal grid: by using an hexagonal grid, the final loss must be weighted based on the grid design (each cell has 6 neighbors and each hexagonal ring has 6 more hexagons than the previous ring). The only component we chose is the form of the distance weight decay (the "inverse" function 1/x): we indeed use the simplest form of weight decay formulation to not add any ad-hoc decay function which would then require, as you correctly pointed out, an ablation study.
> >       -  Hexagonal convolutions:
> >           -  Hexagonal convolutions give better results than standard convolutions in case of hexagonal shaped data, as demonstrated by [hoogeboom2018hexaconv] and [steppa2019hexagdly], so we have not added any ablation study regarding this natural choice. We have added these references in the new submitted paper.
> >       -  Embedding size (bottleneck layer):
> >           -  The embedding size for GeoVeX and Hex2Vec is equal, in order to not provide any advantage based on a potential bigger representation space. We have added this information in the new version of the paper to be more clear. We plan to do a sensitivity analysis of this parameter in the future works: empirically we used the biggest size after which we did not notice any change in the learnt structures nor reconstruction loss.
> >
> > [wang2020isotropic] Wang, Lu, et al. "The isotropic organization of DEM structure and extraction of valley lines using hexagonal grid." Transactions in GIS 24.2 (2020): 483-507.
> >
> > [van1995score] Van den Broek, Jan. "A score test for zero inflation in a Poisson distribution." Biometrics (1995): 738-743.
> >
> > [yang2010score] Yang, Zhao, James W. Hardin, and Cheryl L. Addy. "Score tests for zero-inflation in overdispersed count data." Communications in Statistics—Theory and Methods 39.11 (2010): 2008-2030.
> >
> > [hoogeboom2018hexaconv] Hoogeboom, Emiel, et al. "Hexaconv." arXiv preprint arXiv:1803.02108 (2018).
> >
> > [steppa2019hexagdly] Steppa, Constantin, and Tim L. Holch. "HexagDLy—Processing hexagonally sampled data with CNNs in PyTorch." SoftwareX 9 (2019): 193-198.

---

> > > ### Comment · Reviewer_zeEd · 2022-12-07
> > > **Thanks for the rebuttal**
> > >
> > > Thanks for the clarification on novelty of the paper. However, contextual embedding is a well-studied idea in machine learning domain and is not novel. Thus, I agree with Reviewer 3Ztx that ICLR may not be a suitable venue and won't vote for acceptance of the paper.

---

### Official Review · Reviewer_3Ztx · 2022-10-28

**Confidence:** 4
**Correctness:** 2
**Technical Novelty And Significance:** 2
**Empirical Novelty And Significance:** 2
**Recommendation:** 3

**Clarity, Quality, Novelty And Reproducibility:**

clarity: the paper's basic idea is clear

novelty: using a hexagon convolution encoder seems to be novel, but existing work has already used the same data for learning spatial embeddings and achieved similar performance (Hex2Vec).

quality: more experiments on multi-scale/global spatial data analysis tasks are preferred.

**Strength And Weaknesses:**

S1. The problem is important and using convolutional autoencoders in building global spatial embeddings seems to be novel.

W1. The method is very similar to Hex2Vec (both based on hexagons and OSM data). The performance improvement in the experiments is also very marginal.

W2. The scale effect is very important for spatial big data analysis. However, the current method only learns representation for a fixed scale of the spatial region (H3 hexagon), which may not be useful for practical spatial data analysis.

W3. The evaluation is only conducted for US and Italy, which cannot well support the motivation that the learned embedding is for the whole earth.


**Summary Of The Paper:**

The paper proposes to learn spatial embedding for coordinates based on H3 hexagons and convolutional autoencoders using the OSM’s tag data.

**Summary Of The Review:**

In brief, learning global spatial embedding is a valuable task. However, the currently proposed method and the used data lack significant contribution beyond existing work.

---

> ### Author Response · Authors · 2022-11-18
> **Thank you very much for your valuable review! We have made some changes and we have submitted a new version of the paper!**
>
> Thank you very much for your valuable review! We have made some changes and we have submitted a new version of the paper!
>
> > "S1: The problem is important and using convolutional autoencoders in building global spatial embeddings seems to be novel."
>
> Thank you for acknowledging the novelty of hexagonal convolutional autoencoders with ZIP head for count data!
>
> > "W1: The method is very similar to Hex2Vec (both based on hexagons and OSM data). The performance improvement in the experiments is also very marginal."
>
> - Despite the similarities, there is fundamental difference between GeoVeX and Hex2Vec: Hex2Vec does not consider any information from the neighboring hexagons. As a consequence, Hex2Vec embeddings cannot be considered contextualized in the sense the learnt representation of an hexagon does not change based on where this hexagon is in the world. This can be problematic in terms of semantics as shown in the qualitative comparison between GeoVeX and Hex2Vec embedding vectors for the city of Los Angeles (paragraph A.1 in Appendix of the initial submission). While for GeoVeX, we can clearly recover the coast, city center, etc., there is nothing human-readable for the Hex2Vec vectors since each hexagon's embedding is not influenced by the surrounding hexagons.
>
> - Based on your relevant observations, we understood that the difference was not easy to understand, so we improved the Related Work section and we added a qualitative analysis (Section 4.1) where we integrated more analyses on the semantic captured by GeoVeX vectors and by Hex2Vec vectors. We can clearly see some geographic structures captured by GeoVeX, while we get a lot of noisy latent vectors in Hex2Vec, since not contextualized. You can find this analysis in the new version of the paper!
>
> - "Marginal improvement": it is true it can seem small in absolute terms, but given that the geo embeddings are pre-trained and task-agnostic, we believe that the improvement is interesting. To be more confident about our statements, we ran multiple times the experiments and we added the standard deviation of the results in the tables. Thanks for pointing this out, because by including the range, GeoVeX appears as a winner in a more clear fashion!
>
> > "W2: The scale effect is very important for spatial big data analysis. However, the current method only learns representation for a fixed scale of the spatial region (H3 hexagon), which may not be useful for practical spatial data analysis."
>
> We agree with you, this resolution can be a hyperparameter to tune. However, by using the highest resolution (= the smallest possible H3 hexagons) and by covering all the world, more coarse representation can be easily made by aggregating the embeddings at different scales/polygons (eg a city, a country, a custom shape). Due to our current hardware constraints, the highest H3 resolution on our machines was 8. A similarity here can be made with Word2Vec and Doc2Vec, where Word2Vec represents the highest resolution embedding for a specific word (= the smallest hexagon in our case), and Doc2Vec represents the vector of the set of words composing a document (= a region/polygon in our case).
>
> > "W3: The evaluation is only conducted for US and Italy, which cannot well support the motivation that the learned embedding is for the whole earth."
>
> Thank you for pointing this out! We have downloaded more data from inside-airbnb and we have extended the test of the price-prediction task to cover cities all around the world (see Appendix A.5 for the full list of cities and regions, to allow reproducibility), and the results are showing that GeoVeX is now more clearly improving upon other models. With the extension of the experiments data to worldwide regions, and the addition of a qualitative analysis of the geographic structures learnt by GeoVeX compared to Hex2Vec, the final submission should better convey the improvement we are seeing during the experiments and the exploration!

---

> > ### Comment · Reviewer_3Ztx · 2022-11-21
> > **Thanks for the response**
> >
> > I appreciate the response and new experiments at a global scale. Now I am more clear about the difference between your method and Hex2Vec. However, considering neighboring hexagons seems not a very challenging task from the deep learning perspective. Hence, this paper may still lack the deep technical novelty for a  machine learning conference like ICLR. Maybe the conference on spatial data, e.g., ACM SIGSPATIAL, is more appropriate.

---

### Author Response · Authors · 2022-12-06
**Hello dear Reviewers, thank you very much for your valuable reviews and interesting started discussions! Don't hesitate if you need more clarification or other points to be addressed on our end.**

Hello dear Reviewers, thank you very much for your valuable reviews and interesting started discussions!
Don't hesitate if you need more clarification or other points to be addressed on our end.

---

### Decision · Program_Chairs · 2023-01-20

**Decision:**

Reject

**Justification For Why Not Higher Score:**

The empirical evaluation is limited as highlighted in the metareview. Given this is a primarily empirical work, there needs to be robust ablations, comparisons, and discussions around various design choices.

**Justification For Why Not Lower Score:**

N/A

**Metareview: Summary, Strengths And Weaknesses:**

This paper proposes an algorithm for learning geospatial embeddings based on an autoencoder with hexagonal convolutions. The model is trained using a Poisson loss. Empirical benefits are shown over 2 downstream tasks. The reviewers uniformly agree on the importance of the problem being studied. There is also a mixed acknowledgement of the empirical results in the paper and the novelty --- some reviewers appreciated these aspects whereas others find the paper lacking on both accounts. There was a key question regarding the differences with hex2vec which the authors aptly clarified. The major pending concerns with the paper are on (a) the limited empirical evaluations w.r.t downstream tasks and comparisons with related baselines [sheehan2019predicting] and [mac2019presence]; (b) the strong claims on the generality of OSM in comparison with e.g., Wikipedia; (c) the lack of ablations for the different components of the work (simply citing other works is insufficient). I believe all these concerns are important for full justice to the generality claimed in this work, so I'd encourage the authors to conduct further experiments to address them for a future submission.